# Cannabidiol activates neuronal Kv7 channels

**Han-Xiong Bear Zhang[1†], Laurel Heckman[2†], Zachary Niday[1†], Sooyeon Jo[1†], Akie Fujita[1], Jaehoon Shim[2], Roshan Pandey[2], Hoor Al Jandal[2], Selwyn Jayakar[2], Lee B Barrett[2], Jennifer Smith[3], Clifford J Woolf[1,2]\*, Bruce P Bean[1]\***

[1]Department of Neurobiology, Harvard Medical School, Boston, United States; [2]F.M. Kirby Neurobiology Research Center, Boston Children's Hospital, Boston, United States; [3]ICCB-Longwood Screening Facility and Department of Immunology, Harvard Medical School, Boston, United States

\*For correspondence:
clifford.woolf@childrens.harvard.edu (CJW);
bruce_bean@hms.harvard.edu (BPB)

†These authors contributed equally to this work

**Competing interest:** The authors declare that no competing interests exist.

**Abstract** Cannabidiol (CBD), a chemical found in the *Cannabis sativa* plant, is a clinically effective antiepileptic drug whose mechanism of action is unknown. Using a fluorescence-based thallium flux assay, we performed a large-scale screen and found enhancement of flux through heterologously expressed human Kv7.2/7.3 channels by CBD. Patch-clamp recordings showed that CBD acts at submicromolar concentrations to shift the voltage dependence of Kv7.2/7.3 channels in the hyperpolarizing direction, producing a dramatic enhancement of current at voltages near –50 mV. CBD enhanced native M-current in mouse superior cervical ganglion starting at concentrations of 30 nM and also enhanced M-current in rat hippocampal neurons. The potent enhancement of Kv2/7.3 channels by CBD may contribute to its effectiveness as an antiepileptic drug by reducing neuronal hyperexcitability.

## Editor's evaluation

Cannabidiol (CBD) has attracted great interest as a potential therapy for epilepsies and has been shown to be effective in several syndromic forms of pediatric epilepsy. This study finds that clinically relevant concentrations of CBD enhance neuronal M-current, a potassium current whose activation is antiepileptic. These findings open up the possibility that activation of M-current could underlie antiepileptic efficacy of CBD.

## Introduction

Cannabidiol (CBD), a phytocannabinoid present in marijuana (*Mechoulam et al., 1970*), has been shown in recent clinical trials to be an effective agent for treating some forms of epilepsy in children, including Dravet syndrome (*Devinsky et al., 2017*; *Devinsky et al., 2018b*; *Devinsky et al., 2019*; *Miller et al., 2020*) and Lennox–Gastaut syndrome (*Devinsky et al., 2018a*; *Thiele et al., 2019*). How CBD ameliorates epileptic activity is unclear (*Rosenberg et al., 2015*; *Rosenberg et al., 2017*; *Franco and Perucca, 2019*). Unlike Δ(9)-tetrahydrocannabinol (THC), the other major phytocannabinoid in marijuana, CBD does not activate CB1 or CB2 G-protein-coupled receptors (*Pertwee, 2005*). At micromolar concentrations, CBD has inhibitory effects on a wide range of proteins, including many receptors and channels (*Ibeas Bih et al., 2015*; *Watkins, 2019*). Like many classic antiepileptic agents, CBD inhibits voltage-dependent sodium channels in a state-dependent manner, with reported half-maximal concentrations of ~2–10 µM (*Hill et al., 2014*; *Patel et al., 2016*; *Ghovanloo et al., 2018*; *Mason and Cummins, 2020*). However, as CBD reduction of overall epileptiform activity can be detected in brain slice preparations at much lower concentrations (*Jones et al., 2010*), the

importance of sodium channel inhibition for CBD's anticonvulsant effects remains uncertain (*Hill et al., 2014*). Other molecular targets that could mediate antiepileptic actions of CBD have been described, notably antagonism of the lipid-activated G-protein-coupled receptor GPR55 (*Ryberg et al., 2007*; *Sylantyev et al., 2013*; *Kaplan et al., 2017*), and electrophysiological effects correlated with GPR55 antagonism have been described at concentrations of CBD as low as 200 nM (*Sylantyev et al., 2013*).

The most potent effect of CBD on a well-defined electrophysiological function so far reported is an inhibition of endocannabinoid modulation of synaptic transmission (*Straiker et al., 2018*). This effect of CBD is mediated by a negative allosteric effect on CB1 receptors, with CBD acting at a site distinct from the primary binding site (*Laprairie et al., 2015*). Electrophysiologically, this inhibitory negative allosteric effect is detectable at 100 nM and is substantial at 500 nM (*Straiker et al., 2018*). Here, we report that CBD acts at concentrations as low as 30 nM to activate neuronal M-current, a non-inactivating potassium current mediated by Kv7 channels that activate at subthreshold voltages. CBD shifts the voltage dependence of activation of these channels in the hyperpolarizing direction, resulting in a significant activation of Kv7 current at subthreshold voltages. These results suggest that the activation of neuronal M-current may be one mechanism by which CBD exerts its antiepileptic action.

## Results
### CBD activates heterologously expressed Kv7.2/7.3 channels
We discovered the ability of CBD to activate Kv7.2/7.3 channels in a screen using fluorescence signals from thallium entry evoked by depolarization of a Chinese hamster ovary (CHO) cell line stably expressing human Kv7.2 and Kv7.3 channels. In a screen of a library of 154 compounds chosen from structures with known or possible ion channel modulating activity (*Figure 1—source data 1*), CBD was the only compound to produce a substantial enhancement of the fluorescence signal, except for retigabine and flupirtine, both known activators of Kv7.2/7.3 channels.

We then tested the action of CBD on the Kv7.2/7.3 cell line using whole-cell patch-clamp recordings and saw a dramatic enhancement of the currents activated by depolarization, with particularly large effects for currents activated near –50 mV. *Figure 1A* shows an example, where 100 nM CBD produced a doubling of the current activated at –50 mV, while there was little effect at –20 mV, where channels are near-maximally activated in the control situation. 100 nM enhanced the current evoked at –50 mV by an average factor of 2.8 ± 0.4 (n = 20), while 300 nM CBD enhanced the current by a factor of 4.6 ± 0.5 (n = 14).

The enhancement of the Kv7.2/7.3-mediated current was produced by a shift of the voltage-dependent activation of the channels in the hyperpolarizing direction (*Figure 1C*). In collected results, 300 nM CBD shifted the midpoint for channel activation by an average of –13.9 ± 0.9 mV (n = 17). The shift in the voltage dependence of activation reached a maximum of about –20 mV at CBD concentrations of 3–10 µM, with CBD acting with a half-maximal concentration of about 200 nM (*Figure 1D*).

We next tested whether CBD enhances native Kv7 channels in neurons using measurements of M-current in mouse superior cervical ganglion (SCG) neurons. Using the classic voltage protocol for distinguishing M-current from other potassium currents by virtue of its non-inactivating property and activation at subthreshold voltages (*Brown and Adams, 1980*), we used a steady holding voltage of –30 mV and hyperpolarizing voltage steps to quantify the M-current from its characteristic slow, voltage-dependent deactivation. Application of CBD at concentrations of 30–300 nM produced a dose-dependent enhancement of M-current (*Figure 2*), with enhancement of the steady-state outward current at –30 mV and of the slowly deactivating current seen during hyperpolarization to –60 or –70 mV, a defining characteristic of M-current (*Figure 2A*). It was also clear that CBD shifted the voltage dependence of M-current, resulting in less complete deactivation for a step to –60 mV (*Figure 2A*). In collected results quantifying the effect of CBD, the enhancement of M-current measured at –50 mV increased from a factor of 1.85 ± 0.19 with 30 nm CBD (n = 14) to a factor of 3.02 ± 0.56 with 300 nm CBD (n = 9).

To test whether CBD enhancement of M-current also occurs in central neurons likely involved in epilepsy, we tested CBD on potassium currents in hippocampal neurons (*Figure 3*). To facilitate application of well-defined concentrations of CBD without potential problems from absorption into the bulk tissue of brain slices, we used a preparation of cultured rat hippocampal neurons. Using a voltage

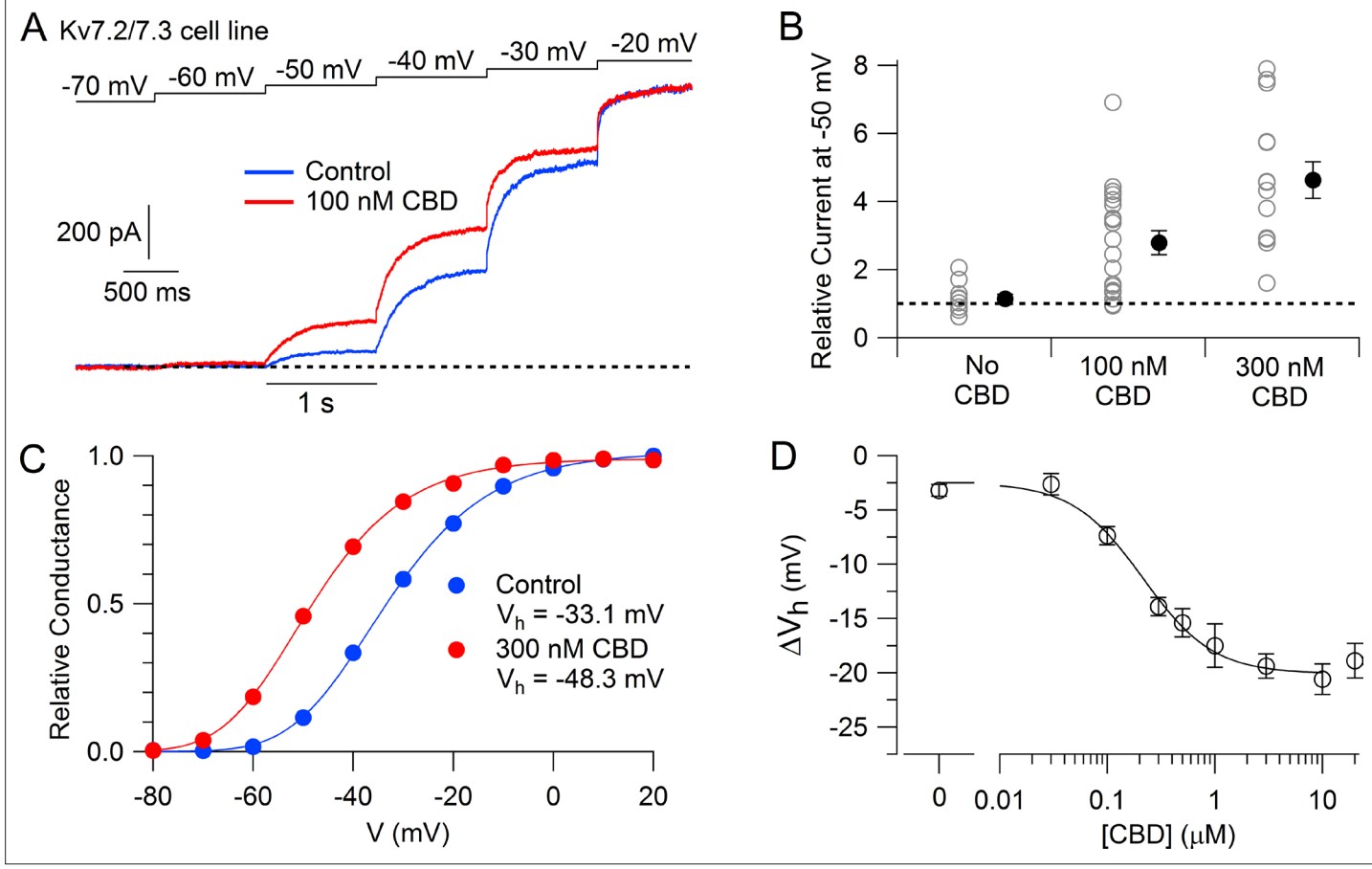

**Figure 1.** Cannabidiol (CBD) enhancement of cloned human Kv7.2/7.3 channel current in Chinese hamster ovary (CHO) cells. (**A**) hKv7.2/7.3 current evoked by staircase depolarizations before and after application of 100 nM CBD. (**B**) Collected results (mean ± SEM) for current at –50 mV after application of 100 nM (n = 20) or 300 nM CBD (n = 14) for 4–6 min, normalized to current before CBD application, using the protocol in (**A**). 'No CBD' values (n = 11) are for 6 min dummy applications of solution containing only vehicle (DMSO). (**C**) Voltage-dependent activation of hKv7.2/7.3 channels measured in a cell before and after application of 300 nM CBD. Relative conductance at each voltage was measured from the initial tail current at a step to –50 mV following 1 s depolarizations to voltages between –100 mV and +20 mV from a holding potential of –80 mV. Solid lines: fits to data points of fourth power Boltzmann function, $[1/(1 + \exp(-(V - V_{hn})/k))]^4$, where V is test pulse voltage, $V_{hn}$ is voltage of half-maximal activation for single 'n' particle, and k is slope factor for activation of n particles. Control: $V_{hn}$ = –54.4 mV, k = 12.8 mV (midpoint of function = –33.1); 300 nM CBD: $V_{hn}$ = –67.9 mV, k = 11.8 mV (midpoint of function –48.3 mV). (**D**) Concentration-dependent shift of activation midpoint by CBD. Measurements of the midpoint were made before and 10 min after exposure to CBD at various concentrations. mean ± SEM, n = 9 for 30 nM CBD, n = 21 for 100 nM CBD, n = 17 for 300 nM CBD, n = 12 for 500 nM CBD, n = 7 for 1 μM CBD, n = 16 for 3 μM CBD, n = 19 for 10 μM CBD, n = 10 for 20 μM CBD. Value for 0 CBD represents the measurement of a small shift that occurred with dummy applications of DMSO-containing control solution for 10 min (n = 11). Solid line: fit to the Hill equation, $\Delta V_h = -2.5$ mV $- 17.5$ mV/(1 + (EC$_{50}$/[CBD])^n$_H$), where EC$_{50}$ = 214 nM and the Hill coefficient n$_H$ = 1.3.

The online version of this article includes the following source data for figure 1:

**Source data 1.** Screen data and source data for *Figure 1*.

protocol designed to emphasize M-current (holding the neurons at –30 mV and stepping to –50 mV), CBD enhanced the outward current at both –30 mV and –50 mV in 16 of the 20 cells tested. Consistent with this action of CBD being an enhancement of M-current, which in hippocampal neurons is mediated by Kv7.2, Kv7.3, and Kv7.5 (*Shah et al., 2002*), there was no increase if CBD was applied in the presence of the Kv7 inhibitor XE-991 (*Wang et al., 1998*; *Brown and Passmore, 2009*). In fact, CBD applied after XE-991 produced on average a small (13% ± 4%, n = 15) decrease in current at –50 mV, consistent with a weak inhibitory effect on other, non-M-currents.

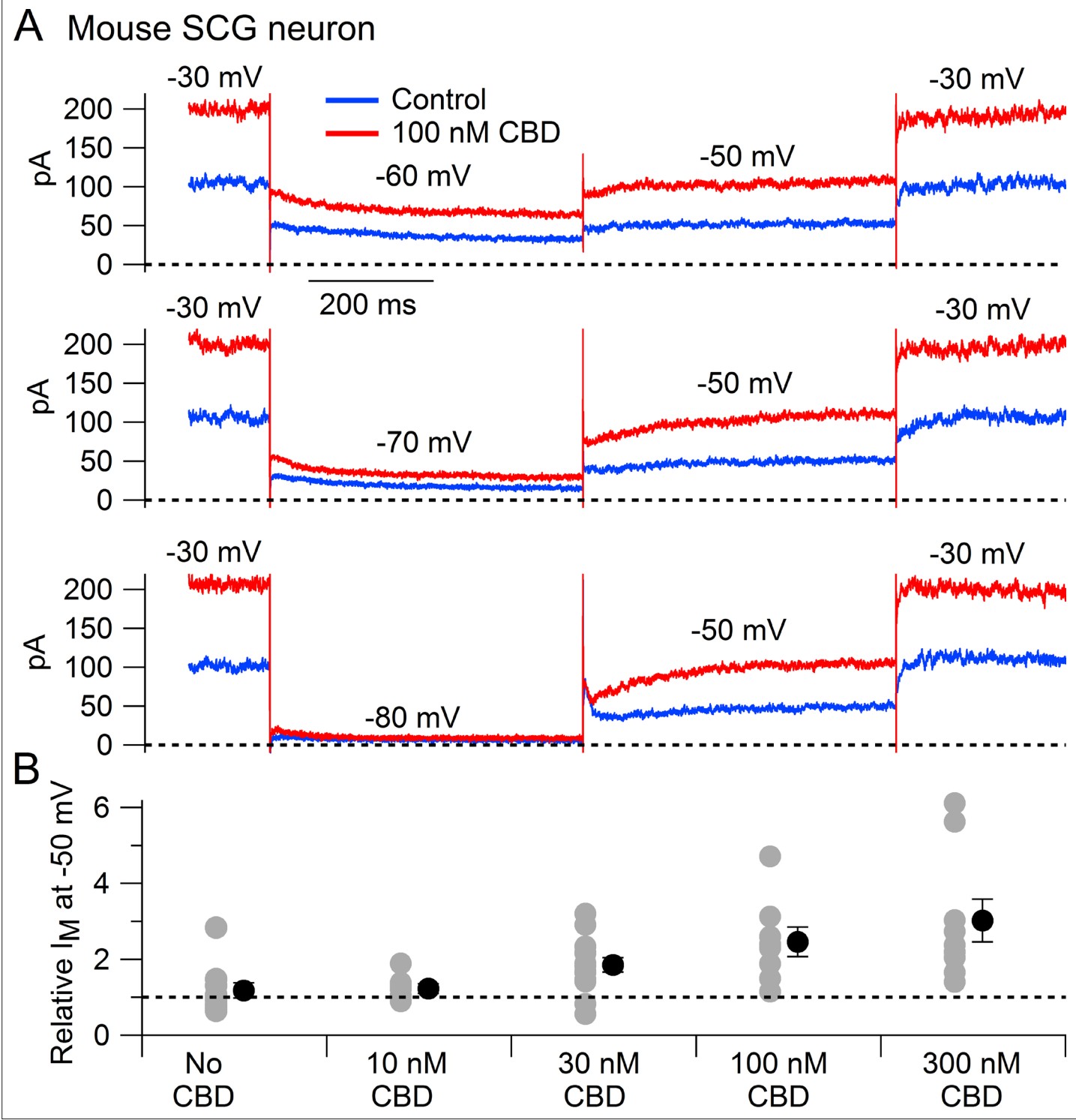

**Figure 2.** Cannabidiol (CBD) enhancement of M-current in mouse sympathetic neurons. (**A**) Currents evoked by hyperpolarizations to –60 mV, –70 mV, and –80 mV from a holding potential of –30 mV before (blue) and after (red) application of 100 nM CBD. (**B**) Collected results (mean ± SEM) for effect of CBD on steady-state M-current at –50 mV. Current was read at the end of a 1 s step from –30 mV to –50 mV, normalized to current before CBD application, following exposure to 10 nM CBD (n = 7), 30 nM CBD (n = 14), 100 nM CBD (n = 8), or 300 nM CBD (n = 9). The maximum effect of CBD was reached in 6–9 min for 10 nM and 30 nM CBD and 2–6 min for 100 nM and 300 nM CBD. 'No CBD' values (n = 10) are for 7–9 min dummy applications. Gray circles: individual cells. Black circles: mean ± SEM. Non-paired two-tailed t-tests: 10 nM CBD vs. No CBD, p=0.85; 30 nM CBD vs. No CBD, p=0.024; 100 nM CBD vs. No CBD, p=0.015; 300 nM CBD vs. No CBD, p=0.012.

*Figure 2 continued on next page*

*Figure 2 continued*

The online version of this article includes the following source data for figure 2:

**Source data 1.** Source data for *Figure 2*.

## Discussion

Kv7 channel-mediated M-current plays a major role in controlling the excitability of many types of neurons, including neocortical pyramidal neurons (*Barrese et al., 2018*; *Brown and Passmore, 2009*; *Gunthorpe et al., 2012*; *Vigil et al., 2020*; *Jepps et al., 2021*). Enhancement of M-current is a clinically proven mechanism of antiepileptic action, as demonstrated by the clinical efficacy of retigabine, an antiepileptic drug that acts by enhancement of current through Kv7 channels (*Wickenden et al., 2000*; *Tatulian et al., 2001*; *Gunthorpe et al., 2012*; *Sills and Rogawski, 2020*). Our results suggest that the clinical efficacy of CBD could result at least in part by the enhancement of the Kv7-mediated M-current in central neurons. As in the case of retigabine, it remains to be determined exactly which

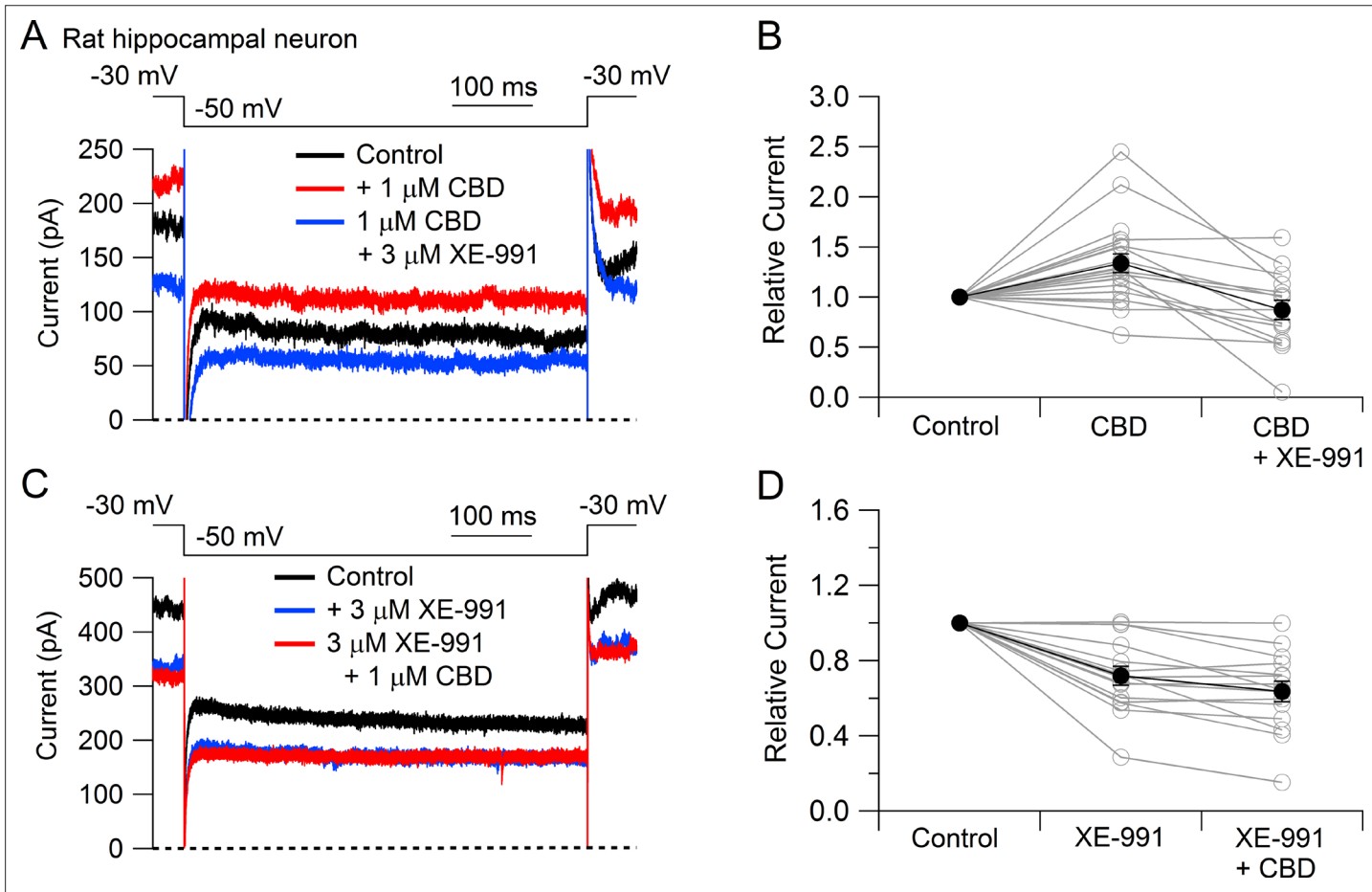

**Figure 3.** Cannabidiol (CBD) enhancement of Kv7 current in rat hippocampal neurons. (**A**) Currents at a holding voltage of –30 mV and during a 500 ms hyperpolarization to –50 mV in control, after application of 1 μM CBD, and after addition of 3 μM XE-991 in the continuing presence of CBD. (**B**) Collected data with this protocol. Current was measured at the end of the step to –50 mV, normalized to current before application of CBD. Connected open circles indicate data for individual cells (n = 20 for application of CBD, n = 15 for application of CBD followed by XE-991) and closed circles represent mean ± SEM. Paired *t*-test for currents after CBD compared to control currents, p=0.00017 (n = 20, two-tailed), paired *t*-test for currents in CBD + XE-991 compared to CBD, p=0.00038 (n = 15, two-tailed). (**C**) Currents in control, after application of 3 μM XE-991, and after addition of 1 μM CBD in the continuing presence of XE-991. (**D**) Collected data with symbols as in (**B**); n = 15 cells for application of XE-991 followed by CBD. Paired *t*-test for currents after XE-991 compared to control, p=0.00071 (n = 15, two-tailed), paired *t*-test for currents in XE-991 + CBD compared to XE-991, p=0.0105 (n = 15, two-tailed).

The online version of this article includes the following source data for figure 3:

**Source data 1.** Source data for *Figure 3*.

populations of neurons are most sensitive to this enhancement of M-current, and how these effects alter the overall network activity relevant to epileptic activity.

Interestingly, the effect of CBD in enhancing the neuronal M-current is the opposite of the effect of cannabinoids that act as agonists at the CB1 receptor, which inhibit M-current in hippocampal neurons (*Schweitzer, 2000*). Thus, the fact that CBD is not a CB1 agonist – and actually acts as an allosteric antagonist at CB1 receptors (*Laprairie et al., 2015*; *Straiker et al., 2018*) – may be an important aspect of its mechanism of action. The opposite effects on M-current of CBD and CB1 agonists like THC fit well with the history of the development of CBD as an antiepileptic drug, which began with anecdotal evidence that extracts from a particular strain of cannabis with high CBD and low THC ('Charlotte's Web') were an effective adjunctive therapy for a child with Dravet syndrome (*Maa and Figi, 2014*; *Rosenberg et al., 2015*; *Williams and Stephens, 2020*).

In doing our experiments, an important step was discovering that use of plastic containers and plastic tubing could greatly reduce the apparent effects of solutions with CBD at submicromolar concentrations, likely reflecting loss of CBD by absorption into plastic as occurs with other cannabinoids with similarly high lipophilicity (*Christophersen, 1986*; *Hippalgaonkar et al., 2011*). This issue complicates a comparison of the concentration dependence with which CBD affects various targets potentially relevant for its antiepileptic activity. Although published data for CBD inhibition of sodium channels have typically reported half-blocking concentrations of 2–10 µM, we have found that when using glass reservoirs and tubing to apply well-defined concentrations of CBD to isolated neurons, substantial inhibition of steady-state 'persistent' sodium current can be seen with 30–100 nM CBD (unpublished results). CBD at 100 nM also significantly depresses endocannabinoid modulation of synaptic transmission (*Straiker et al., 2018*), suggesting that the overall effects of submicromolar concentrations of CBD on neuronal excitability could involve multiple actions. Further work will be required to evaluate the relative importance of actions on various targets to CBD's antiepileptic action.

Our results add to recent experiments demonstrating that Kv7.2/7.3 channels are susceptible to enhancement by a wide variety of agents acting by several different mechanisms (*De Silva et al., 2018*; *Manville and Abbott, 2018c*; *Manville and Abbott, 2018a*; *Miceli et al., 2018*; *Wang et al., 2018*; *Kanyo et al., 2020*; *Kurata, 2020*; *Li et al., 2020*). Such agents include endogenous compounds like GABA (*Manville et al., 2018b*), the ketone body β-hydroxybutyrate (*Manville et al., 2020*), and arachidonic acid metabolites and derivatives (*Schweitzer et al., 1990*; *Schweitzer et al., 1993*; *Larsson et al., 2020a*; *Larsson et al., 2020b*), as well as a variety of natural products including cilantro (*Manville and Abbott, 2019*). Further development of Kv7.2/7.3 enhancers for treating epilepsy and other neuronal disorders seems promising (*Maljevic and Lerche, 2014*; *Vigil et al., 2020*), especially because retigabine has been withdrawn from clinical use because of a number of off-target side effects (*Brickel et al., 2020*). Compared to other compounds recently found to enhance Kv7.2/7.3 channels, CBD has the distinction of having already been successfully used in multiple epilepsy clinical trials. However, CBD is far from a perfect drug (*Sekar and Pack, 2019*) as it requires large dosages and has a complex pharmacokinetic profile that limit its effective oral administration (*Millar et al., 2019*). Improved knowledge of CBD's most important molecular targets should allow for the design of novel compounds that retain its key molecular actions but with improved pharmacokinetics and reduced off-target effects.

## Materials and methods

### Key resources table

| Reagent type (species) or resource | Designation | Source or reference | Identifiers | Additional information |
|---|---|---|---|---|
| Strain, strain background (*Mus musculus*) | Swiss Webster | Charles River | Cat# 024 | |
| Strain, strain background (*Rattus norvegicus*) | Sprague–Dawley | Charles River | Cat# 400 | |

*Continued on next page*

Continued

| Reagent type (species) or resource | Designation | Source or reference | Identifiers | Additional information |
|---|---|---|---|---|
| Cell line (*Cricetulus griseus*) | Kv7.2/7.3 CHO cell line | Mayflower Bioscience | BSYS-KV7.2/3-CHO-C | CHO (Chinese hamster [*C. griseus*] ovary) cell line stably transfected with recombinant human Kv7.2/7.3 ion channels |
| Commercial assay or kit | MycoAlert PLUS Mycoplasma Detection Kit | Lonza | LT07-703 | |
| Commercial assay or kit | FluxOR II Green Potassium Ion Channel Assay | Invitrogen | LT07-703 | |
| Chemical compound, drug | Cannabidiol | Cayman Chemical | Cat# 90080, CAS 13956-29-1 | |
| Chemical compound, drug | Ham's F12-Glutamax-l medium | Gibco | Cat# 31765-035 | |
| Chemical compound, drug | Penicillin-streptomycin | Gibco | Cat# 15140-122 | |
| Chemical compound, drug | Puromycin | InvivoGen | Cat# ant-pr-1 | |
| Chemical compound, drug | Papain | Worthington Biochemical | Cat# LS003126 | |
| Chemical compound, drug | L-15 | Gibco | Cat# 11415-064 | |
| Chemical compound, drug | Neurobasal A Medium | Gibco | Cat# 10888-022 | |
| Chemical compound, drug | B-27 | Gibco | Cat# 17504-010 | |
| Chemical compound, drug | Penicillin-streptomycin | Sigma-Aldrich | Cat# P4333 | |
| Chemical compound, drug | Minimal Essential Medium | American Tissue Type Collection | Cat# DMEM 30-2002 | |
| Chemical compound, drug | Hank's Balanced Salt Solution | Gibco | Cat# 14170-112 | |
| Chemical compound, drug | DMEM/F12 | Gibco | Cat# 11330-032 | |
| Chemical compound, drug | Tetrodotoxin w/citrate | Abcam | Ab120055 | |
| Software, algorithm | Clampex | Molecular Devices | Version 10.3.1.5 | https://www.moleculardevices.com |
| Software, algorithm | Igor Pro | WaveMetrics | Version 6.12A | https://www.wavemetrics.com |
| Software, algorithm | DataAccess | Bruxton Corporation | | http://www.bruxton.com/DataAccess/index.html |

## Thallium flux assay

### Cell culture

CHO cells coexpressing human Kv7.2 and Kv7.3 channels (Mayflower Bioscience, BSYS-KV7.2/3-CHO-C) were cultured at 37°C in 5% $CO_2$ in a Thermo Scientific incubator in Ham's F12-Glutamax-l medium (Gibco, Cat# 31765-035) supplemented with 10% fetal bovine serum (Gibco), 1% penicillin/streptomycin solution (Gibco, Cat# 15140-122), and 5 µg/mL puromycin (InvivoGen, Cat# ant-pr-1). The cell line was validated by patch-clamp recording of large voltage-activated currents (>1 nA for depolarizations to 0 mV) that reversed at the potassium equilibrium potential, had the voltage dependence and kinetics previously reported for Kv7.2/7.3 heteromeric channels expressed in CHO cells (*Tatulian et al., 2001*), and were enhanced by 3 µM retigabine (*Tatulian et al., 2001*). The cell line was tested for mycoplasma contamination using the Lonza MycoAlert PLUS Mycoplasma Detection Kit (LT07-703, Lonza Pharma & Biotech). Cells were seeded in 15 cm dishes at 200,000 cells per dish, fed twice weekly, and cultivated once weekly. 24 hr before the start of the screen, the culture dishes were trypsinized, and a Countess automated cell counter (Invitrogen) was used to quantify cell numbers before plating them into four Greiner poly-D-lysine-coated 384-well black clear-bottomed microplates at 20,000 cells per well in 40 µL media using a Multidrop Combi Reagent Dispenser. The four microplates were incubated overnight in a Thermo Scientific incubator at 37°C in 90% humidity and 5% $CO_2$.

## Compound preparation and handling

The Panacea Channel Modulator Library, a custom collection of 154 compounds oriented toward known or possible ion channel modulators, was assembled and deposited at the ICCB-Longwood Screening Facility, Harvard Medical School. Compound metadata are listed in *Figure 1—source data 1* (assay raw data). Each compound was assayed at four concentrations. The compounds were initially plated as stock solutions in DMSO at concentrations of 0.08 mM, 0.4 mM, 2 mM, and 10 mM, which yielded final assay concentrations of 267 nM, 1.3 µM, 6.7 µM, and 33 µM. Using a custom Seiko compound transfer workstation, 300 nL of experimental compound stock solutions, as well as positive (retigabine at 10 mM in DMSO) and negative (DMSO) controls, were pin transferred into a Greiner Bio-One 384 Deep Well Small Volume polypropylene microplate containing 30 µL of 1× FluxOR chloride-free buffer. This resulted in 16 positive and 16 negative control wells on every assay plate. Each of the two compound microplates was screened in duplicate (four assay plates).

## Kv7.2/7.3 assay

The FluxOR potassium channel assay (Thermo Fisher) was performed using a Hamamatsu FDSS 7000 plate reader essentially as outlined in the product sheet. After the Kv7.2/7.3 CHO cells were incubated in four 384-well assay microplates for 24 hr, a 40 mL solution of FluxOR dye was made by combining 400 µL Powerload concentrate (100×), 40 µL of 13 FluxOR II green reagent (1000× fluorescent dye) in DMSO, 31.2 mL purified water, 4 mL 10× FluxOR assay buffer, 4 mL FluxOR II background suppressor, and 400 µL probenecid (100× in water). Next, media were aspirated from each well of the assay microplates containing Kv7.2/7.3 CHO cells using an Agilent Bravo Liquid Handling system. The assay microplates were then washed two times with FluxOR chloride-free buffer diluted from 5× to 1× (20 µL per well per wash). After the second wash was removed, 7.68 mL of the 40 mL dye solution was dispensed to each 384-well assay microplate (20 µL per well). The assay microplates were incubated in the dye solution at room temperature protected from light for 45 min. Subsequently, 10 µL of diluted compounds in FluxOR chloride-free buffer were added to each assay microplate from the compound dilution plate prepared as described above, resulting in final compound concentrations of 267 nM, 1.3 µM, 6.7 µM, and 33 µM. Assay microplates were incubated in compound and dye for 15 min at room temperature protected from light. For the assay, stimulus buffer was first prepared by mixing 50 mM thallium sulfate ($Tl_2SO_4$, 4.8 mL), FluxOR chloride-free buffer (5×, 6.0 mL), and purified water (19.2 mL). Next, 19.2 mL of this stimulus solution (50 µL per well) was loaded into an additional Greiner Bio-One 384 Deep Well Small Volume polypropylene microplate. The four assay plates and plate containing stimulus buffer were then loaded onto a Hamamatsu FDSS 7000Ex plate reader and liquid handler. For each assay microplate, 10 µL of stimulus buffer was added per well after 50 s for a final concentration of 4 mM Tl+ in the assay plate. Fluorescence was measured for 600 data points

(~3 min) at 4 Hz. FDSSv3.3.1 software was used for baseline correction and data analysis. All results from the screen are shown in *Figure 1—source data 1* (assay raw data).

## Electrophysiology with CHO Kv7.2/7.3 cell line

Cells were maintained and passaged in a humidified 37°C incubator in sterile culture flasks containing Ham's F12-Glutamax-l medium (Gibco, Cat# 31765-035) supplemented with 10% fetal bovine serum (Gibco), 1% penicillin/streptomycin solution (Gibco, Cat# 15140-122), and 5 µg/mL puromycin (InvivoGen, Cat#ant-pr-1), and cells were passaged at a confluence of about 50–80%. For electrophysiological recordings, cells were seeded onto 12 mm cover slips (Fisherbrand, Cat# 12-545-80). Whole-cell patch-clamp recordings were made using a Multiclamp 700B Amplifier (Molecular Devices). Electrodes were pulled from borosilicate capillaries (VWR International, Cat# 53432-921) on a Sutter P-97 puller (Sutter Instruments), and shanks were wrapped with Parafilm (American National Can Company) to allow optimal series resistance compensation without oscillation. The resistances of the pipettes were 1.8–3.5 MΩ when filled with the intracellular solution consisting of 140 mm KCl, 10 mM NaCl, 2 mM MgCl$_2$, 1 mm EGTA, 0.2 mm CaCl$_2$, 10 mM HEPES, 14 mM creatine phosphate (Tris salt), 4 mM MgATP, and 0.3 mM GTP (Tris salt), pH adjusted to 7.4 with KOH. Seals were formed in Tyrode's solution consisting of 155 mM NaCl, 3.5 mM KCl, 1.5 mM CaCl$_2$, 1 mM MgCl$_2$, 10 mM HEPES, 10 mM glucose, pH 7.4 adjusted with NaOH. After establishing whole-cell recording, cell capacitance was nulled and series resistance was partially (~70%) compensated. The cell was then lifted and placed in front of an array of quartz fiber flow pipes (250 µm internal diameter, 350 µm external diameter, Polymicro Technologies, Cat# TSG250350) attached with styrene butadiene glue (Amazing Goop, Eclectic Products) to a rectangular aluminum rod (cross section 1.5 cm × 0.5 cm) whose temperature was controlled by resistive heating elements and a feedback-controlled temperature controller (Warner Instruments, TC-344B). Solutions were changed (in ~1 s) by moving the cell from one pipe to another. Recordings were made at 37°C.

Voltage commands were delivered and current signals were recorded using a Digidata 1321A data acquisition system (Molecular Devices) controlled by pCLAMP 10.3 software (Molecular Devices). Current and voltage records were filtered at 5 kHz and digitized at 100 kHz. Analysis was performed with Igor Pro 6.12 (WaveMetrics, Lake Oswego, OR) using DataAccess (Bruxton Software) to import pClamp data.

The effects of CBD on Kv7 current in the cell line were quantified in two ways: by the enhancement of current evoked at –50 mV during stair-step protocols like that in *Figure 1A* and by the shift in midpoint of activation curves as in *Figure 1C*. In both cases, current records were corrected for linear capacitive and leak current by subtracting scaled responses to signal-averaged 5 mV hyperpolarizations delivered from –80 mV. Calculation of the enhancement of current at –50 mV during the stair-step protocol was confined to cells in which the current was at least 20 pA in control in order to minimize any error resulting from imperfect leak correction. For determining activation curves, the voltage dependence of activation was measured from the initial tail current at a step to –50 mV following 1 s depolarizations to voltages between –100 mV and +40 mV from a holding potential of –80 mV. Tail current was averaged over a 1 ms interval starting at a time when the immediate jump in current had settled, typically 0.8–1.6 ms after the voltage step. Plots of normalized tail current versus test voltage could be fit well by a Boltzmann function raised to the fourth power. The midpoint of activation was measured in a fit-independent manner by calculating the test voltage at which tail current reached half of its maximal value (reached at voltages between 0 to +40 mV) using linear interpolation between the test voltages straddling the midpoint. Calculation of shifts of activation midpoint by CBD was confined to cells in which the maximal tail current at –50 mV was at least 100 pA and in which the activation curve in CBD was fit well by a Boltzmann function raised to the fourth power.

CBD (Cayman Chemical, Cat# 90080, CAS 13956-29-1) was prepared as a 10 mM stock solution in DMSO, which was diluted in the external Tyrode's solution to the final concentration. DMSO was added to the control solution at the same concentration as in the CBD solution. In early experiments, CBD-containing solutions were prepared in polystyrene test tubes and applied to cells from reservoirs made from 10 mM polypropylene syringe bodies. Realizing that phytocannabinoids have exceptionally high lipophilicity (*Thomas et al., 1990*) and can apparently partition into plastic (*Christophersen, 1986*; *Hippalgaonkar et al., 2011*), we then switched to using glass reservoirs from which solutions flowed through hollow quartz fibers to be applied to cells. We found that using glass reservoirs and

tubing resulted in larger and more reproducible effects of CBD concentrations of 1 µM and below. The reported data for these concentrations are confined to experiments using glass reservoirs and tubing. The effects of concentrations of 3 µM and above were not less when using plastic reservoirs, and the collected data for concentrations of 3–20 µM include experiments done with both plastic and glass reservoirs.

## Preparation of SCG neurons

SCG were removed from adult Swiss Webster mice of either sex (postnatal day 56), cut in half, and treated for 20 min at 37°C with 20 U/mL papain (Worthington Biochemical, Cat# LS003126) in a calcium- and magnesium-free (CMF) Hank's buffer (Gibco, Cat# 14170-112) containing 137 mM NaCl, 5.36 mM KCl, 0.33 mM $Na_2HP_4$, 0.44 mM $KH_2PO_4$, 4.2 mM $NaHCO_3$, 5.55 mM glucose, and 0.03 mM phenol red. The ganglia were then treated for 20 min at 37°C with 3 mg/mL collagenase (type I; Roche Diagnostics, Cat# 10103586001) and 4 mg/mL Dispase II (Roche Diagnostics, Cat# 37045800) in CMF Hank's buffer. Cells were dispersed by trituration with a fire-polished glass Pasteur pipette in a solution composed of two media combined in a 1:1 ratio: Leibovitz's L-15 medium (Gibco, Cat# 11415-064) supplemented with 5 mM HEPES and DMEM/F12 medium (Gibco, Cat# 11330-032) and plated onto coverslips. Then cells were incubated at 37°C (5% $CO_2$) for 2 hr, after which Neurobasal medium (Gibco, Cat# 10888-022) containing B-27 supplement (Gibco, Cat# A3582801), and penicillin and streptomycin (Sigma-Aldrich, Cat# P4333) was added to the dishes. Cells were stored at room temperature and used within 48 hr.

## Electrophysiology with SCG neurons

Whole-cell patch-clamp recordings were made using a Multiclamp 700B Amplifier (Molecular Devices) interfaced to a Digidata 1321A data acquisition system (Molecular Devices) controlled by pCLAMP 10.3 software (Molecular Devices). Electrodes were 2–4 MΩ when filled with the intracellular solution consisting of 140 mM K aspartate, 13.5 mM NaCl, 1.6 mM $MgCl_2$, 5 mM EGTA, 9 mM HEPES, 14 mM creatine phosphate (Tris salt), 4 mM MgATP, 0.3 mM Tris-GTP, pH 7.2 adjusted with KOH, with shanks wrapped with Parafilm to allow optimal series resistance compensation (70–80%). Seals were formed in Tyrode's solution consisting of 155 mM NaCl, 3.5 mM KCl, 1.5 mM $CaCl_2$, 1 mM $MgCl_2$, 10 mM HEPES, 10 mM glucose, pH 7.4 adjusted with NaOH, and cells were lifted in front of quartz fiber flow pipes attached to a temperature-controlled aluminum rod. M-current was recorded with external Tyrode's solution containing 1 µM TTX and 10 µM $CdCl_2$ and quantified by measuring the current at the end of a 1 s step to –50 mV from a steady holding potential of –30 mV, after subtracting linear leak current determined by extrapolation of current measured at voltages between –80 mV and –90 mV. However, the traces in *Figure 2A* show raw records with no correction of capacitive current or leak current. Recordings were made at 37°C.

Voltage commands were delivered and current signals were recorded using a Digidata 1321A data acquisition system (Molecular Devices) controlled by pCLAMP 10.3 software (Molecular Devices). Current and voltage records were filtered at 5 kHz and digitized at 50 kHz. For display, current records were smoothed by binomial (Gaussian) smoothing using a smooth factor of 101 sampling intervals, equivalent to low-pass filtering with a time constant of about 80 µs. Analysis was performed with Igor Pro 6.12 (WaveMetrics) using DataAccess (Bruxton Software) to import pClamp data.

## Preparation of rat hippocampal neurons

Primary cultures of hippocampal neurons were prepared from rat embryos (E19–E20). Pregnant female Sprague–Dawley rats were anesthetized with isoflurane. The skin was washed with 70% ethanol, the peritoneal cavity was opened, and embryos were transferred into ice-cold preparation solution $Ca^{2+}/Mg^{2+}$-free HBSS (Gibco, Cat# 14170-112) with 5 mM HEPES (Gibco, Cat# 15630-080) and 1 mM sodium pyruvate (Gibco, Cat# 11360-070) in a 100 mm Petri dish on ice. Heads and brains were sequentially dissected from embryos, with the ice-cold preparation solution exchanged during each step. Under a dissecting microscope, the meninges were stripped away from the cerebral hemispheres and dorsal hippocampi were dissected with a fine scissor. The hippocampal pieces were transferred into a pre-warmed preparation solution containing 37U papain (Worthington, Cat# LS003126), 5 mM L-cysteine (Sigma, Cat# C7352), and 1080U DNase I (Sigma, Cat# DN-25), incubated at 37°C for 15 min, and then washed three times with enzyme-free warmed preparation solution. The preparation solution

was then exchanged for a titration medium (EMEM, ATCC, Cat# 30-2003), 5% FBS (Gibco, Ca# 16140-071), and 1× penicillin/streptomycin (P/S, Gibco, Cat# 15140-122), and the hippocampal pieces were titrated using Pasteur pipettes fire-polished to two different tip sizes. After determining cell density using a hematocytometer, a maintenance medium (Neurobasal media [Gibco, Cat# 21103-049], 2% B27 [Gibco, Cat# 17504-044], 5 mM glutamine [Gibco Cat# 25030-081], and 1× P/S) was added into cell suspension to make cell density of $1–1.5 × 10^5$/mL. Five poly-D-lysine (Sigma, Cat# P-7405)-coated coverslips (Fisherbrand, Cat# 12-545-80) were placed in 35 mm dishes and $2–3 × 10^5$ cells were plated in each 35 mm dish ($≥4–6 × 10^4$ cells/coverslip). Neurons were maintained for 13–17 days in vitro (DIV). Every 2–3 days, half of the medium was removed from the 35 mm dishes and replaced with the same volume of the fresh maintenance solution.

All experiments using animals were performed according to an institutional IACUC-approved protocol.

## Electrophysiology with rat hippocampal neurons

Recordings were made from neurons after 13–17 DIV. Neurons with three processes and a pyramidal shape were selected for recording. To avoid problems arising from absorption of CBD to plasticware, recordings were made in an all-glass chamber made by attaching a glass ring (18 mm outer diameter, 3 mm height, Thomas Scientific 6705R24) to a glass-bottom microwell dish (MatTek# P35G-1.5-20C). Whole-cell recordings were obtained using patch pipettes with resistances of 2.2–2.5 MΩ when filled with the internal solution, consisting of 140 mM K-gluconate, 9 mM NaCl, 1.8 mM $MgCl_2$, 0.09 mM EGTA, 9 mM HEPES, 14 mM creatine phosphate (Tris salt), 4 mM MgATP, and 0.3 mM Tris-GTP, pH adjusted to 7.2 with KOH. The shank of electrode was wrapped with Parafilm to allow optimal series resistance compensation. Seals were obtained and the whole-cell configuration established in Tyrode's solution consisting of 155 NaCl, 3.5 KCl, 1.5 $CaCl_2$, 1 $MgCl_2$, 10 HEPES, 10 glucose, pH adjusted to 7.4 with NaOH, with added 1 μM TTX. Reported membrane potentials are corrected for a liquid junction potential of –13 mV between the K-gluconate-based internal solution and the Tyrode's solution in which current was zeroed at the start of the experiment. The amplifier was tuned for partial compensation of series resistance (typically 40–70% of a total series resistance of 4–10 MΩ), and tuning was periodically readjusted during the experiment. Currents were recorded with a Multiclamp 700B Amplifier (Molecular Devices), filtered at 5 kHz with a low-pass Bessel filter, and digitized using a Digidata 1322 A data acquisition interface controlled by pCLAMP 9.2 software (Molecular Devices). Recordings were made at 30°C.

M-current was evoked by 500 ms steps to –50 mV from a steady holding potential of –30 mV. Stock solutions of 10 mM CBD in DMSO and 20 mM XE-991 in DMSO were made in glass vials and diluted into Tyrode's solution (in glass vials) as 20 μM CBD or 60 μM XE-991 on the day of recording. Aliquots of these solutions were applied directly into the glass chamber and mixed with a 100 μL pipettor to make final concentrations of 1 μM CBD or 3 μM XE-991, respectively. To minimize any residual effect of CBD from the previous recording, the glass chamber was rinsed with 70% ethanol for three times and distilled water for three times before putting a new coverslip into the chamber.

## Acknowledgements

This work was supported by the NIH (NS105076, NS36855, NS110860, U54HD090255), DARPA (HR0011-19-2-0022), and the Charles R Broderick III Phytocannabinoid Research Initiative. We are grateful to Dr. Mustafa Sahin and Ms. Candace Tong-Li of the Assay Development and Screening Core Facility of the Boston Children's Hospital Intellectual & Developmental Disabilities Research Center for support and help.

# Additional information

## Funding

| Funder | Grant reference number | Author |
| --- | --- | --- |
| National Institute of Neurological Disorders and Stroke | NS36855 | Bruce P Bean |
| National Institute of Neurological Disorders and Stroke | NS110860 | Bruce P Bean |
| National Institute of Neurological Disorders and Stroke | NS105076 | Clifford J Woolf |
| Eunice Kennedy Shriver National Institute of Child Health and Human Development | U54HD090255 | Clifford J Woolf |
| Defense Advanced Research Projects Agency | HR0011-19-2-0022 | Clifford J Woolf Bruce P Bean |
| Charles R. Broderick III Phytocannabinoid Research Initiative | | Bruce P Bean Clifford J Woolf |

The funders had no role in study design, data collection and interpretation, or the decision to submit the work for publication.

## Author contributions

Han-Xiong Bear Zhang, Laurel Heckman, Zachary Niday, Sooyeon Jo, Jaehoon Shim, Roshan Pandey, Jennifer Smith, Conceptualization, Data curation, Formal analysis, Investigation, Methodology, Writing – review and editing; Akie Fujita, Conceptualization, Data curation, Formal analysis, Methodology, Writing – review and editing; Hoor Al Jandal, Data curation, Formal analysis, Investigation, Writing – review and editing; Selwyn Jayakar, Conceptualization, Data curation, Formal analysis, Investigation, Methodology, Project administration, Supervision, Writing – review and editing; Lee B Barrett, Data curation, Methodology, Project administration, Supervision, Validation; Clifford J Woolf, Conceptualization, Funding acquisition, Methodology, Project administration, Writing – review and editing; Bruce P Bean, Conceptualization, Data curation, Formal analysis, Funding acquisition, Methodology, Project administration, Supervision, Writing – original draft, Writing – review and editing

## Author ORCIDs

Laurel Heckman (iD) http://orcid.org/0000-0003-0124-0519
Bruce P Bean (iD) http://orcid.org/0000-0002-5093-3576

## Ethics

Animal experimentation: This study was performed in strict accordance with the recommendations in the Guide for the Care and Use of Laboratory Animals of the National Institutes of Health. All of the animals were handled according to approved Institutional Animal Care and Use Committee (IACUC) protocols of Harvard Medical School ( Protocol 02538).

## Decision letter and Author response

Decision letter https://doi.org/10.7554/eLife.73246.sa1
Author response https://doi.org/10.7554/eLife.73246.sa2

# Additional files

## Supplementary files
• Transparent reporting form

## Data availability

Source data is provided for data in all figures, in the Source Data files for each figure.

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
