## [Editor Report]

Cannabidiol (CBD) has attracted great interest as a potential therapy for epilepsies and has been shown to be effective in several syndromic forms of pediatric epilepsy. This study finds that clinically relevant concentrations of CBD enhance neuronal M-current, a potassium current whose activation is antiepileptic. These findings open up the possibility that activation of M-current could underlie anti-epileptic efficacy of CBD.

---

## [Decision Letter]

**Decision letter after peer review:**

Thank you for submitting your article "Cannabidiol activates neuronal Kv7 channels" for consideration by *eLife*. Your article has been reviewed by 3 peer reviewers, including Jon T Sack as the Reviewing Editor and Reviewer #1, and the evaluation has been overseen by Kenton Swartz as the Senior Editor. The following individual involved in review of your submission has agreed to reveal their identity: Ken Mackie (Reviewer #2).

Essential revisions:

We think you have made a finding important for the fields of M-current, epilepsy, and CBD research, that can be effectively communicated as an *eLife* Short Report. We imagine the significance of your finding could be enhanced by addressing:

1) The CBD EC50 for M-current in hippocampal neurons.

2) CBD impacts on action potential firing in hippocampal neurons.

3) Whether CBD impacts on Nav currents at low concentrations which are efficacious against M-current (e.g., 100 nM) with the same solution delivery system.

A straightforward way to address such revision could be through inclusion of a few key additional results while keeping the scope of the study limited and consistent with a Short Report.

We look forward to a revised manuscript, and also to see what future research reveals with CBD and M-current.

*Reviewer #1:*

This research presents clear evidence that cannabidiol activates Kv7.2/7.3 channels and neuronal M-current. A strength of this work is the finding that CBD activates Kv7 channels a heterologous system and endogenous M-currents (presumptive Kv7 channels) in neurons from rats and mice. The observation of consistent effects suggests across cell-types and species that this result is robust and will translate to Kv7 M-current in other cell types. Another strength is the identification of the basic mechanism by which CBD acts: shifting the voltage dependence of Kv7 current activation to more negative voltages. This study finds that 100 nM concentration of CBD can activate Kv7 M-current, a concentration that is reported to be ineffective against most other CBD-modulated proteins, though a direct comparison of effective concentrations against other CBD targets is not tested here. Results are interpreted with appropriately nuanced discussion considering the promiscuous effects of CBD on membrane proteins and the range of other molecules that modulate Kv7 channels.

A claim of this manuscript is that CBD acts at lower concentrations on Kv7 than it acts on its many other targets. As nicely described in the methods, careful solution handling is needed to observe sensitivity of Kv7 channels to 100 nM CBD. I noted that DRG neuron Nav current is inhibited by 300 nM CBD in your recent study. Clarifying how distinct the concentration-response of Kv7 and Nav currents are with the same solution handling seems important to substantiate this claim.

*Reviewer #2:*

The strengths of this study are its identification of a novel potential mechanism for the anti-epileptic actions of CBD, the evaluation across four modalities/cell types (thallium flux, patch clamp in CHO cells over expressing Kv7.2/Kv7.3, mouse super cervical ganglion (SCG) neurons, and cultured mouse hippocampal neurons), and the finding that M current is activated by concentrations of CBD that are similar to those achieved in the plasma of patients administered CBD as an anti-epileptic. The experiences of the authors with plastic versus glass and the adsorption of CBD will be very helpful for others following up on their studies.

The only limitation of this study is that it doesn't explore the mechanism(s) for CBD enhancement of M current (e.g., interactions with PIP2, etc.), however the primary finding of this study is so significant for the field and it will undoubtedly stimulate much additional experimentation, that this is a minor limitation and doesn't negatively affect the impact of the work presented here.

1. Page 3: CBD does affect perception (e.g., https://pubmed.ncbi.nlm.nih.gov/32247649/ ), so it is incorrect to say that it is not psychoactive. It certainly lacks the characteristic psychoactivity of THC.

2. Page 3: "direct primary ligand" is confusing. Would say it's not an orthosteric ligand as some work suggests that there is defined allosteric binding site on CB1 for CBD.

3. Page 5, top: Should "M-current activation" by "M-current deactivation"?

4. Page 12, in the paragraph describing compound preparation, a number of concentrations on my pdf were mM where they should probably be uM.

*Reviewer #3:*

The manuscript investigated the effects of cannabidiol on KV7.2/KV7.3 channel activity in cell lines and neurons. The authors show that cannabidiol enhances expressed KV7.2/KV7.3 channel activity in cell line at very low doses with a half maximal concentration for activating the current at -50 mV being approximately 200 nM. As KV7.2/KV7.3 channels encode for the M-current in neurons, they also tested the effects of cannabidiol on the M-current in cultured rat SCG neurons and hippocampal neurons. Cannabidiol at 300 nM enhanced the M-current in SCG neurons. However, 1 μm cannabidiol produced a much smaller effect at enhancing the M-current in cultured rat hippocampal neurons (Figure 3).

Overall, the effects of cannabidiol on the M-current are interesting and this may be the potential mechanism of action by which cannabidiol exerts its' anti-epileptic effects. The study is, though, very limited and could be extended to include the effects of cannabidiol on neuronal activity. It might also be interesting to test whether compounds with similar structures to cannabidiol also enhance the M-current.

1) The authors state that they tested a library of 154 compounds chosen from the 'structures with known or possible ion channel modulating activity' (page 4). Could the authors please expand on where they got this library from and which compounds this library contained?

2) The lesser effect of cannabidiol on the M-current in cultured rat hippocampal neurons compared with SCG neurons suggests that the EC50 for cannabidiol on the M-current in hippocampal neurons differs from that in SCG neurons. The authors ought to determine the EC50 values for the compound on the M-current in hippocampal neurons and SCG neurons. If they differ, the authors should attempt to explain why this might be the case.

3) If the EC50 values for cannabidiol differ in hippocampal neurons compared with that obtained for expressed KV7.2/KV7.3 currents in cell lines or the M-current in SCG neurons, it raises questions on whether cannabidiol may exert its anti-epileptic effects via activation of the M-current. Thus, the authors should investigate whether low concentrations of cannabidiol (100 nM) that have been reported to reduce epileptiform activity in brain slices reduce hippocampal neuronal action potential firing and if so, whether this is by activating the M-current. This would be a really important and essential experiment to determine if cannabidiol exerts its anti-epileptic effect by enhancing the M-current.

---

## [Author Response]

Reviewer #1:[…] A claim of this manuscript is that CBD acts at lower concentrations on Kv7 than it acts on its many other targets. As nicely described in the methods, careful solution handling is needed to observe sensitivity of Kv7 channels to 100 nM CBD. I noted that DRG neuron Nav current is inhibited by 300 nM CBD in your recent study. Clarifying how distinct the concentration-response of Kv7 and Nav currents are with the same solution handling seems important to substantiate this claim.

The reviewer makes a very good point. After submission of the manuscript, we have been doing further experiments on different kinds of sodium channels in a variety of preparations using glass reservoirs and tubing. We have found that CBD inhibits different kinds of sodium channels with different potency. The inhibition of sodium channels is strongly state-dependent, so the concentration-response is complicated and dependent on the voltage protocol. Most relevant for potential anti-epileptic action, we see inhibition of subthreshold persistent sodium current in native central neurons (predominantly carried by Nav1.6 channels) by concentrations similar to those that enhance M-current, with substantial effects at CBD concentrations as low as 30 nM in both cases. Therefore, we currently think it likely that the anti-epileptic effects of CBD could involve both M-current enhancement and state-dependent sodium channel inhibition. In the revised paper, we include new data on native M-current in sympathetic neurons with lower concentrations of CBD than we studied initially. The new data demonstrates enhancement by 30 nM CBD. And, we have added a sentence to the Discussion mentioning our unpublished data showing that 30 nM CBD also produces some inhibition of persistent sodium current in some central neurons. We note that submicromolar concentrations of CBD also significantly depress endocannabinoid modulation of synaptic transmission (Straiker et al., 2018), suggesting that overall effects of submicromolar concentrations of CBD on neuronal excitability may well involve multiple actions.

Reviewer #2:[…] 1. Page 3: CBD does affect perception (e.g., https://pubmed.ncbi.nlm.nih.gov/32247649/ ), so it is incorrect to say that it is not psychoactive. It certainly lacks the characteristic psychoactivity of THC.

Thanks, good point – we have removed the statement that it is not psychoactive.

2. Page 3: "direct primary ligand" is confusing. Would say it's not an orthosteric ligand as some work suggests that there is defined allosteric binding site on CB1 for CBD.

Thanks – we have changed the wording to simply say that is not an activator of CB1 or CB2 receptors.

3. Page 5, top: Should "M-current activation" by "M-current deactivation"?

Thanks, the wording was clumsy. The terminology is tricky because M-current is activated by depolarization but the experimental measurements are done using deactivation. It seems simplest to just say that CBD alters the voltage-dependence of M-current so we have used that terminology.

4. Page 12, in the paragraph describing compound preparation, a number of concentrations on my pdf were mM where they should probably be uM.

Thanks! Those concentrations referred to the stock solution concentrations in the plate of compounds that were then diluted by 300-fold when applied to the assay plate, but that was not clear. We have modified this section to be clearer about the dilutions from the stock solutions to the final concentrations in the experimental wells. We have also made this clear in the description of file of raw data from the assay that we added to Source Data associated with Figure 1.

Reviewer #3:[…]1) The authors state that they tested a library of 154 compounds chosen from the 'structures with known or possible ion channel modulating activity' (page 4). Could the authors please expand on where they got this library from and which compounds this library contained?

We have now added a spreadsheet from the screen (as part of Source Data for Figure 1) that includes the list of compounds, along with sources of the compounds, concentrations that were tested, and the raw data from the thallium flux fluorescence measurements for each compound.

2) The lesser effect of cannabidiol on the M-current in cultured rat hippocampal neurons compared with SCG neurons suggests that the EC50 for cannabidiol on the M-current in hippocampal neurons differs from that in SCG neurons. The authors ought to determine the EC50 values for the compound on the M-current in hippocampal neurons and SCG neurons. If they differ, the authors should attempt to explain why this might be the case.

We have spent a good part of the four months since receiving the reviews attempting to better define the dose-response of CBD in both SCG neurons and hippocampal neurons.

In SCG neurons, we originally studied a concentration of 300 nM because it gave a near maximal-effect in the experiments on cloned channels. To define the dose-response of native channels, we have now done new experiments with 10, 30 and 100 nM CBD, along with dummy applications of CBD-free solutions. The data show a clear enhancement of native M-current by 30 nM CBD and increasing effects as the CBD concentration increases. We have added these new data to Figure 2. However, the data do not allow a clear determination of an EC_50_ value. Unlike the cloned channels where we could define the midpoint of activation and use this as a measure of effect, in the neurons we found it impossible to define complete G-V curves because of the difficulty of distinguishing M-current from other currents at voltages positive to -30 mV. We spent several weeks experimenting with various blocker cocktails to inhibit other channels but were unable to sufficiently block other channels activated positive to -30 mV to enable quantification of pure M-current at these voltages. Therefore, we were unable to quantify the concentration-dependence of CBD’s action by the value of the shift of midpoint, as we did for the cloned channels. We believe the best procedure is to display all of the data for all the cells at each concentration, as we have now done in Figure 2B. At all concentrations of CBD, there is substantial cell-to-cell variability in the degree of enhancement, which we believe may reflect different degrees of activation of the channels at -50 mV in the basal condition. Despite this cell-to-cell variability, the collected results show clearly that 30 nM CBD enhances the current and there is a clear increase of effect with higher CBD concentrations.

Quantification in hippocampal neurons was even more difficult. The reason we did experiments with hippocampal neurons in culture instead of in slice is that in early experiments studying excitability of hippocampal CA1 neurons in brain slice experiments, we found that concentrations of CBD at concentrations as high as 2 μm had almost no effect on firing when applied to slices, even though they strongly inhibit firing of acutely-isolated neurons. We believe this is because the tissue in the slice soaks up the highly lipophilic CBD and prevents effective application to the cell being recorded from, even at levels of several uM. We note that published experiments with CBD in brain slice preparations have generally used far higher concentrations of CBD to see electrophysiological effects, very likely reflecting this problem. We turned to experiments with cultured hippocampal neurons because this allowed application of well-defined CBD concentrations. However, a limitation of cultured neurons is that there a large cell-to-cell variability in the size of the XE-991 sensitive current, and some cells have none. You can see this variability in the data in Figure 3B and 3D, where you can see that a significant fraction of cells had very little or no current inhibited by XE-991. An even bigger challenge with the cultured neurons was obtaining whole-cell recordings that were stable for more than ~ 5-10 minutes before there were changes in leak current. With low concentrations of CBD, it takes ~7-8 minutes for the effects to reach steady-state. A third limitation of quantifying effects on M-current in hippocampal neurons, even in brain slice recordings of M-current under control conditions, is that because the channels are in the axon initial segment, the currents recorded in the soma are relatively small and often have their apparent voltage-dependence shifted in the depolarizing direction (e.g. Yamada-Hanff and Bean, J Neurosci. 33:15011-15021, 2013) so that to separate them from other currents it is necessary to use XE-991, as we did (Figure 3). However, recovery from XE-991 is too slow to allow an “XE-991-subtraction” quantification of both control and CBD-enhanced current, which is why we had to do experiments with a separate population of neurons to test whether XE-991 prevented the enhancing effect of CBD (Figure 3C,D).

Since the reviews, we have tried hard to develop a preparation of hippocampal or cortical neurons that would allow application of low concentrations of CBD for long enough to quantify their effects on M-current, but this has been unsuccessful. We spent several weeks doing experiments with acutely-dissociated hippocampal pyramidal neurons. These allow robust recordings with little change in leak current, but we found that the acutely-isolated hippocampal neurons have little or no M-current, probably because of loss of the axon. We also tried a variety of culture conditions and longer culture times with both mouse and rat hippocampal cultures and also mouse cortical cultures but were ultimately unable to find a preparation where recordings of M-current lasted long enough to allow studies with CBD below 1 uM. So far, the most promising preparation is a preparation of cultured cortical neurons differentiated from human induced pluripotent cells, which appear to have substantial M current based on retigabine (and CBD) responses in multi-electrode array recordings, but as the differentiation procedure has a turn-around time of 8 weeks, it will be many months before we can hope to obtain detailed dose-response data with whole-cell recordings, even if the preparation allows good quantification of M-current with somatic recordings – which is still uncertain since the channels may be primarily in axons.

Because the channels underlying most native M-current in both sympathetic neurons and central neurons are likely Kv7.2/7.3 channels, we think it is very likely that the characteristics of CBD enhancement in sympathetic neurons provide a good model for effects on central neurons. However, we agree this has not yet been shown directly and are careful not to imply that it has.

3) If the EC50 values for cannabidiol differ in hippocampal neurons compared with that obtained for expressed KV7.2/KV7.3 currents in cell lines or the M-current in SCG neurons, it raises questions on whether cannabidiol may exert its anti-epileptic effects via activation of the M-current. Thus, the authors should investigate whether low concentrations of cannabidiol (100 nM) that have been reported to reduce epileptiform activity in brain slices reduce hippocampal neuronal action potential firing and if so, whether this is by activating the M-current. This would be a really important and essential experiment to determine if cannabidiol exerts its anti-epileptic effect by enhancing the M-current.

We agree that our data leave open the question of how important activation of M-current is for the overall effect of CBD on epileptic activity. Since our new experiments show activation of native M-current in SCG neurons at 30 nM CBD, we think it is justified to at least suggest that this could be one mechanism of CBD’s anti-epileptic effect. However, as noted above, since submitting the manuscript we have done experiments using glass reservoirs and tubing to study effects of CBD on various kinds of sodium channels and have found that 30 nM CBD can significantly inhibit persistent sodium current in central neurons. Therefore our current working hypothesis is that CBD’s efficacy for epilepsy may reflect combined actions to inhibit sodium current (particularly persistent sodium current and perhaps resurgent sodium current) and to activate M-current. Indeed, these two actions simultaneously would seem ideal for inhibiting firing of cortical pyramidal neurons, where spikes are initiated in the axon initial segment where there is a large persistent sodium current from Nav1.6 channels and also high expression of Kv7.2/7.3 channels that strongly control spike generation. We currently do not have a good experimental preparation that allows dissection of the relative importance of the two mechanisms. We are studying effects on sodium current in acutely dissociated Purkinje and hippocampal neurons, where we can study effects of CBD at submicromolar concentrations, but these acutely dissociated neurons do not have M-current. The cultured hippocampal neurons have M-currents but have small sodium currents and feeble action potential firing compared to slice or acutely dissociated neurons, so they are not well-suited for looking at changes in excitability. We plan to try to develop an all-glass method for local application of CBD in brain slice preparations but we expect it will be challenging with low concentrations of CBD, which take 5-10 minutes to approach steady-state even when applied directly to an acutely isolated cell body with no other tissue around.

In writing the revised version, we have been careful to avoid any implication that activation of M-current is necessarily the only mechanism of CBD’s anti-epileptic action and have added a mention of our on-going experiments showing that low concentrations of CBD also inhibit persistent sodium current..